# Comparing the PRISMA-7 and a Modified Version (PRISMA-6) for Frailty Screening: Addressing Sex Bias in Community-Dwelling Older Adults

**DOI:** 10.3390/geriatrics10010009

**Published:** 2025-01-07

**Authors:** Dietmar Ausserhofer, Angelika Mahlknecht, Verena Barbieri, Adolf Engl, Giuliano Piccoliori, Christian J. Wiedermann

**Affiliations:** Institute of General Practice and Public Health, Claudiana—College of Health Professions, 39100 Bolzano, Italy

**Keywords:** frailty screening, PRISMA-7, sex bias, South Tyrol, population-level screening, primary care, PRISMA-6, older adults

## Abstract

**Background/Objectives:** Frailty screening facilitates the identification of older adults at risk of adverse health outcomes. The Program of Research to Integrate Services for the Maintenance of Autonomy 7 (PRISMA-7) is a widely utilised frailty tool; however, concerns regarding its potential sex bias persist due to item 2, which assigns a frailty point for male sex. This study compared the PRISMA-7 with a modified version, the PRISMA-6 (excluding item 2), to assess their suitability for frailty screening in South Tyrol, Italy. Objectives included evaluating the impact of item 2 on frailty classification and exploring the feasibility of the PRISMA-6 as a more equitable alternative. **Methods:** A cross-sectional survey of 1695 community-dwelling older adults aged ≥75 years was conducted in South Tyrol. Frailty was assessed using both the PRISMA-7 and PRISMA-6. Sociodemographic, health, and lifestyle data were collected to examine associations with frailty classifications. Logistic regression was applied to identify predictors of frailty for each tool. Agreement between the PRISMA-7 and PRISMA-6 was assessed, and internal consistency was evaluated using Cronbach’s alpha. **Results:** Frailty prevalence was 33.9% with the PRISMA-7 and 27.0% with the PRISMA-6. The PRISMA-7 classified men as frail more frequently than women (34.7% vs. 33.0%), while the PRISMA-6 reversed this trend (men, 21.4%; women, 33.0%). Excluding item 2 improved internal consistency (Cronbach’s alpha: PRISMA-7, 0.64; PRISMA-6, 0.75) and aligned frailty classifications with predictors such as age, health status, and physical activity. Logistic regression revealed significant sex differences with the PRISMA-7 but not with the PRISMA-6. **Conclusions:** The PRISMA-7 introduces sex bias by overestimating frailty in men, whereas the PRISMA-6 provides a more equitable and consistent alternative. The findings highlight the PRISMA-6’s potential as a reliable tool for unbiased frailty screening. Future research should validate the PRISMA-6 against established frailty tools to support its integration into primary care settings.

## 1. Introduction

Frailty is a clinical syndrome characterised by a decline in physiological reserves, leading to increased vulnerability to adverse health outcomes, such as falls, hospitalisation, disability, and mortality [1]. As the population ages, identifying frailty in older adults becomes crucial for implementing preventive and management strategies to maintain autonomy and improve the quality of life [2].

The concept of frailty encompasses multiple domains, including physical, psychological, and social factors [3]. Early detection through screening in primary care settings enables timely intervention that may delay or reverse frailty progression [4]. Various screening tools have been developed, among which the Program of Research to Integrate Services for the Maintenance of Autonomy 7 (PRISMA-7) is notable for its brevity and ease of administration [5].

The PRISMA-7 is recommended for use in various healthcare settings [6], including primary care [7,8]; however, significant concerns exist about its limitations, particularly the potential sex bias introduced by item 2, which assigns a frailty point for being male. This item has been criticised for disproportionately identifying men as frail, as shown in validation studies from Brazil [9], China [10], and Turkey [11]. These studies indicate that item 2 exhibits weak correlations with other items, minimally contributes to frailty identification, and lowers the tool’s internal consistency. Excluding item 2 has been found to improve reliability, highlighting the need to address this bias when using the PRISMA-7 in diverse populations. These findings collectively raise concerns about the appropriateness of including item 2 in diverse populations, as it may not align well with the core purpose of the PRISMA-7 tool in detecting frailty or functional loss.

The PRISMA-7 is a seven-item questionnaire. Each item is answered with ‘yes’ or ‘no’, with a score of ≥3 indicating frailty. These questions addressed age, health status, physical capability, and social support.

Italy’s healthcare system is undergoing significant reforms to enhance primary care and community health services. Ministerial Decree No. 77, issued on 23 May 2022, delineates novel organisational models and standards for territorial healthcare assistance [12]. A salient aspect of this decree is the emphasis on the early detection and management of frailty within the community, promoting integrated care pathways, and reinforcing the role of primary care providers [13].

Frailty outcomes are deeply influenced by cultural and societal factors, which shape lifestyle behaviours, health perceptions, and access to care [14,15]. South Tyrol, a multilingual and multicultural region, offers a unique context where German-speaking and Italian-speaking communities coexist under shared healthcare policies but with differing cultural norms. These variations may affect frailty prevalence and the interpretation of sex-based differences. Understanding these influences is essential for designing screening tools like the PRISMA-7 and PRISMA-6 that are equitable and applicable across diverse populations.

This study aimed to evaluate the suitability of the PRISMA-7 as a self-reported frailty-screening tool for community-dwelling individuals aged 75 years and above in South Tyrol’s primary care settings. Specifically, it seeks to assess the impact of item 2 on frailty classification and explore potential sex biases, providing evidence-based recommendations for its implementation considering Ministerial Decree 77/2022.

## 2. Materials and Methods

### 2.1. Study Design

This study was designed as a cross-sectional, population-based survey conducted in the Autonomous Province of Bolzano/South Tyrol, Italy, over a three-month period between 1 March and 30 May 2023. The survey adhered to STROBE guidelines for observational studies, ensuring methodological transparency and rigour [16].

### 2.2. Study Setting and Population

This investigation was conducted in South Tyrol, a multilingual and multicultural region in Northern Italy, where approximately 70% of the population is German-speaking, 25% Italian-speaking, and 5% Ladin-speaking [17]. The target population comprised community-dwelling individuals aged 75 years and older, representing approximately 51,000 older adults in South Tyrol. A stratified probabilistic sampling approach was employed to ensure that the results were representative of the regional population. Sampling accounted for key demographic factors, including age group (75–84 years and ≥85 years), sex (male and female), and residency status (urban and rural communities). The Provincial Institute of Statistics (ASTAT) randomly selected 3600 individuals from the official resident register.

### 2.3. Inclusion and Exclusion Criteria

Eligible participants were community-dwelling older adults aged ≥75 years. Individuals residing in long-term care facilities were excluded from this study, as were those who were unable to complete the questionnaire independently or with the assistance of a family member because of severe health or cognitive impairments.

### 2.4. Data Collection

Data were collected using a standardised questionnaire made available in the three primary languages spoken in the region: German, Italian, and Ladin. Participants were offered three modes of survey completion to accommodate the varying levels of digital and physical accessibility. These comprised self-completion of an online survey hosted on LimeSurvey, paper-based completion with return by post, or telephone interviews administered by trained ASTAT personnel. Non-respondents were followed up with a second invitation one month later to maximise participation.

This investigation achieved a 47% response rate from a stratified probabilistic sample of community-dwelling older adults aged 75 years or older, selected to ensure representativeness across age, gender, and rural–urban residence. While no direct comparison between respondents and non-respondents was conducted, the stratified sampling design aimed to minimise selection bias. This approach ensures that the study sample reflects the broader older population of South Tyrol.

### 2.5. Measures

Frailty was assessed using the PRISMA-7 questionnaire [6], a seven-item instrument wherein responses are coded as binary (yes/no), and a score of three or greater indicates frailty [6] in its German [18] and Italian [19] versions, respectively. To investigate potential sex bias, a modified version of the questionnaire, referred to as the PRISMA-6, was also analysed. The PRISMA-6 excluded item 2. Item 2 of the PRISMA-7 tool, which assigns a frailty point for male sex, was originally included based on evidence suggesting that men may experience frailty differently than women. Research has indicated that older men often face earlier and more severe physical and health-related declines, leading to higher risks of adverse outcomes such as hospitalisation and mortality. As a result, item 2 was intended to serve as a proxy for identifying men who might be at elevated risk [6]. However, subsequent validation studies have questioned the item’s contribution to accurately predicting frailty, suggesting that it may introduce bias rather than enhance diagnostic accuracy. This study critically evaluates the inclusion of item 2 to assess whether its exclusion in the PRISMA-6 results in a more equitable and reliable frailty-screening tool.

Additional sociodemographic data were collected, including age, sex, native language (German, Italian, Ladin, or others), citizenship, education level (below high school or high school and higher), financial resources (excellent, good, adequate, or insufficient), and living situation (alone or with a partner/family). Self-reported health and lifestyle factors were also included, such as perceived health status (poor, moderate, good, or very good). Physical activity was evaluated based on participants’ self-reported weekly engagement and categorised into three levels: two hours or more per week, less than two hours per week, or no physical activity (never). This measure provides an indication of the participants’ engagement in regular physical movement and is considered a key health-related variable in assessing frailty and general well-being. Care utilisation data were collected to assess informal and formal care support, including the use of family assistance, community nursing, private care providers, meals-on-wheels, and frequency of general practitioner (GP) visits.

### 2.6. Statistical Analysis

Given the absence of a gold standard for frailty classification, an indirect assessment strategy was employed to evaluate the performance of the PRISMA-7 and PRISMA-6. Descriptive statistics, including means, medians, and frequencies, were used to summarise the characteristics of the study sample and frailty prevalence based on both instruments. The prevalence of frailty was further stratified by key variables, including sex, age group, education level, and urban or rural residency.

Comparative analyses between the PRISMA-7 and PRISMA-6 were conducted to investigate the influence of item 2. Differences in frailty classification were analysed using the chi-square test for categorical variables and the Wilcoxon rank-sum test for continuous variables. The agreement between the PRISMA-7 and PRISMA-6 was evaluated using proportional overlap, and misclassification rates were determined by identifying individuals classified as frail by one instrument but not the other.

Logistic regression models were employed to identify predictors of frailty, while adjusting for sociodemographic and health-related covariates. Sex-stratified analyses were performed to examine whether the predictors of frailty differed between men and women. Internal consistency for the PRISMA-7 and PRISMA-6 was assessed using Cronbach’s alpha to determine the reliability of the instruments and evaluate the impact of excluding item 2 on overall coherence.

Finally, proxy indicators of frailty, such as age, physical activity levels, perceived health status, and care dependency, were analysed to indirectly validate the performance of the PRISMA-7 and PRISMA-6.

## 3. Results

### 3.1. Sample Characteristics and Frailty Prevalence

This study included 1695 community-dwelling older adults aged 75 years and older. Frailty prevalence varied depending on the scoring tool used. Employing the PRISMA-7, which included item 2 (male sex), 33.9% (*n* = 574) of the participants were classified as frail (Table 1). Conversely, the PRISMA-6, which excluded item 2, identified 27.0% (*n* = 457) as frail (Table 2).

A significant disparity emerged in the frailty prevalence according to sex. The PRISMA-7 indicated a higher frailty prevalence in men (34.7%) than in women (33.0%), whereas the PRISMA-6 reversed this trend, demonstrating a higher frailty prevalence in women (33.0%) than in men (21.4%; *p* < 0.001) (Figure 1).

Age was a robust predictor of frailty, with individuals aged ≥85 years exhibiting significantly higher frailty prevalence than those aged 75–84 years. Using the PRISMA-7, 63.2% of those aged ≥85 years were frail compared to 13.7% in the younger cohort (*p* < 0.001). Similar trends were observed for the PRISMA-6, where frailty prevalence among those aged ≥85 years was 51.4% compared with 10.1% in the 75–84 age group.

### 3.2. Sociodemographic Factors and Frailty

Several sociodemographic variables were associated with frailty (Table 1 and Table 2). The native tongue exhibited a trend towards statistical significance, with German speakers presenting a higher frailty prevalence than Italian speakers, although this disparity diminished when utilising the PRISMA-6. Financial resources emerged as a significant predictor, with individuals reporting insufficient or low financial resources exhibiting a markedly higher prevalence of frailty (PRISMA-7, 46.4%; PRISMA-6, 42.1%; *p <* 0.001).

Living situation did not demonstrate a significant association with frailty in the PRISMA-7 or PRISMA-6 classifications. However, participants with lower educational attainment (below high school) exhibited a significantly higher prevalence of frailty, particularly when assessed using the PRISMA-6. For instance, under the PRISMA-6, 30.0% of individuals with low education levels were classified as frail, compared to 16.2% among those with higher education levels (*p* < 0.001).

### 3.3. Health, Physical Activity, and Frailty

Health status and physical activity demonstrated a strong association with frailty irrespective of the scoring tool used (Table 1 and Table 2). Participants who reported poor or moderate health exhibited substantially higher frailty rates (PRISMA-7: 49.3%; PRISMA-6: 41.1%; *p* < 0.001) than those who reported good or very good health. Similarly, physical inactivity was also a significant predictor. Among participants who did not engage in physical activity, the frailty prevalence was 65.7% in the PRISMA-7 and 60.8% in the PRISMA-6, whereas those who exercised ≥2 h weekly exhibited markedly lower frailty rates (PRISMA-7: 15.6%; PRISMA-6: 9.4%; *p* < 0.001).

### 3.4. Sex Differences in Frailty Under the PRISMA-6

A more comprehensive analysis of the PRISMA-6 results (excluding item 2) revealed significant sex disparities. Frail women demonstrated lower financial resources, with 40.1% of frail women reporting insufficient income compared with 26.6% of frail men (*p* = 0.01). Furthermore, women were more likely to reside alone (54.6%) than frail men (19.7%; *p* < 0.001) and have lower educational attainment. Conversely, frail men exhibited higher levels of physical inactivity, with 32.4% reporting no physical activity than 41.3% of frail women. Table 3 presents the results.

### 3.5. Regression Analysis

Logistic regression models identified predictors of frailty for both the PRISMA-7 and PRISMA-6. The results are presented in Table 4. Using the PRISMA-7, male sex was a significant predictor of frailty (OR = 0.42, 95% CI: 0.31–0.57, *p* < 0.001), reflecting the influence of item 2. However, when employing the PRISMA-6, the effect of sex was no longer significant (OR = 1.34, 95% CI: 0.98–1.83, *p =* 0.067), and other variables assumed greater prominence.

For both instruments, age ≥ 85 years (PRISMA-7: OR = 12.6; PRISMA-6: OR = 9.87; *p* < 0.001), poor financial resources (PRISMA-7: OR = 1.68; PRISMA-6: OR = 2.07; *p <* 0.001), and physical inactivity (PRISMA-7: OR = 6.67; PRISMA-6: OR = 7.58; *p <* 0.001) were strong predictors of frailty. Participants reporting good or very good health were significantly less likely to be classified as frail (PRISMA-7: OR = 0.12; PRISMA-6: OR = 0.11; *p* < 0.001).

### 3.6. Impact of Male Sex on Frailty Classification

The exclusion of item 2 from the PRISMA-7 (PRISMA-6) led to a reduction in the overall frailty prevalence from 33.9% to 27.0%. This shift disproportionately affected men, reducing their frailty prevalence from 34.7% to 21.4%, whereas the prevalence of frailty in women remained largely unchanged at 33.0%. Internal consistency upon the exclusion of item 2 improved (Cronbach’s alpha for the PRISMA-7 = 0.64 vs. the PRISMA-6 = 0.75), suggesting its removal.

## 4. Discussion

This study evaluated the suitability of the PRISMA-7 as a self-reported frailty-screening tool in community-dwelling older adults in South Tyrol, with a particular focus on the impact of item 2 (“Are you male?”) for frailty classification. The findings reveal that while the PRISMA-7 is known to effectively identify frailty, it disproportionately classifies men as frail due to the inclusion of item 2, introducing a potential sex bias. When item 2 was excluded (PRISMA-6), frailty prevalence shifted significantly, with women demonstrating higher frailty rates, aligning with the broader literature on sex differences in frailty. These results challenge the validity of the PRISMA-7 in contexts where sex equity is paramount and suggest that the PRISMA-6 may provide a more accurate and equitable assessment of frailty.

Our findings corroborate prior evidence on sex differences in frailty and extend the discussion to the implications of excluding item 6 from the PRISMA-7 questionnaire, as highlighted in the Green study, where its irrelevance in the Greek context prompted its removal to improve cultural sensitivity and tool applicability [20]. Numerous studies have demonstrated that women exhibit a higher frailty prevalence due to longer life expectancy, greater multimorbidity, and higher rates of disability [21,22,23,24,25]. Furthermore, women experience more severe frailty trajectories and worse outcomes when frail, suggesting differences in frailty profiles rather than in prevalence [26].

The PRISMA-7, with item 2, contradicts these established patterns by consistently identifying a higher prevalence of frailty in men. Validation studies in Brazil, China, and Turkey have similarly highlighted this anomaly, attributing it to item 2’s disproportionate weighting rather than to clinically significant differences in frailty [9,10,11]. These findings, consistent with our results, indicate that the inclusion of item 2 biases frailty classification, particularly in male-dominated or sex-balanced cohorts. The improved internal consistency observed in the PRISMA-6 (Cronbach’s alpha 0.75 compared to 0.64 for the PRISMA-7) is a significant indicator of its reliability as a screening tool. Furthermore, the PRISMA-6 aligns more closely with established predictors of frailty, such as age, health status, and physical activity, enhancing its clinical relevance. The removal of item 2 in the PRISMA-6 eliminates the overclassification of frailty in men observed with the PRISMA-7, which we posit makes it a more balanced and equitable tool. This gender-neutral approach reduces the risk of misallocating resources, ensuring that both men and women are accurately identified for interventions based on their actual frailty levels rather than demographic factors alone.

The implications of the PRISMA-6 extend beyond improved internal consistency. A more accurate and unbiased frailty classification system can lead to better resource allocation and targeted clinical interventions, particularly in primary care settings where frailty screening plays a pivotal role in guiding personalised care plans. While further external validation of the PRISMA-6 is necessary, the results of this study provide a strong rationale for its consideration as a more equitable alternative to the PRISMA-7 in diverse populations.

The exclusion of item 2 in the PRISMA-6 rectified this sex bias, revealing a frailty prevalence pattern more congruent with other clinical and epidemiological assessments. For instance, women demonstrated a higher frailty prevalence under the PRISMA-6, consistent with age-related trends and self-reported health limitations (Table 3). This supports the hypothesis that item 2 artificially inflates frailty prevalence in men, thereby skewing the overall results.

The findings have significant implications for the implementation of frailty screening at the population level, particularly in primary care settings, as mandated by Italian Ministerial Decree 77/2022 [13] and performed in the Italian region of Friuli Venezia Giulia [27]. The PRISMA-7’s apparent sex bias poses a challenge for equitable care delivery. Systematic misclassification of frailty in men could lead to overutilisation of healthcare resources or misallocation of interventions, while underreporting frailty in women could result in missed opportunities for early intervention and prevention [28].

The PRISMA-6, which excludes item 2, has emerged as a more balanced alternative, providing a tool that is better aligned with an evidence-based understanding of frailty [21,22,23,24,25]. Its adoption would mitigate sex bias and allow for more accurate identification of at-risk individuals, particularly women, whose frailty profiles are often underappreciated in male-weighted scoring systems.

Despite the strengths of this study, including its representative population sample and comprehensive analysis, several limitations should be considered. Firstly, a notable limitation of this study is the absence of comparison with other frailty assessment tools that are inherently insensitive to gender influences. While the PRISMA-6 demonstrates improvements in reducing sex bias compared to the PRISMA-7, the inclusion of a validated tool with a gender-neutral design could have further strengthened the analysis and provided additional context for evaluating the performance of the PRISMA-6. The absence of a validated gold standard for frailty, such as the Clinical Frailty Scale or Fried’s Frailty Phenotype, precludes direct sensitivity and specificity analyses of the PRISMA-7 and PRISMA-6 [29]. Second, the study design limits the ability to assess the long-term predictive validity of the PRISMA-7 and PRISMA-6 for adverse health outcomes, such as hospitalisation or mortality. Another limitation of this study is its reliance on self-assessment data, potentially introducing bias, particularly in older participants with cognitive impairments. Cognitive decline, prevalent in individuals aged 75 and older, may affect the accuracy of self-reported frailty questionnaire responses. The absence of data on participants’ cognitive status impedes our evaluation of cognitive impairment’s influence on the results. Lastly, the findings are specific to South Tyrol’s multilingual and multicultural population and may not be generalisable to other settings. Future research should address these limitations by validating the PRISMA-6 against a recognised gold standard in longitudinal studies. Cross-cultural adaptations and validation in diverse populations would also ensure the broader applicability of the findings.

South Tyrol’s sociocultural context provides an opportunity to examine frailty among distinct language groups and cultural traditions. The German- and Italian-speaking communities, despite sharing a healthcare system, may exhibit different health behaviours, attitudes toward ageing, and perceptions of frailty, affecting self-reported outcomes and responses to the PRISMA-7 and PRISMA-6 tools. Language nuances in questionnaire translations could influence interpretations and responses, introducing potential biases in frailty classification. Healthcare access also varies, especially between urban and rural areas, with rural residents facing barriers like longer travel distances to healthcare facilities, potentially worsening frailty due to delayed or limited access to preventive care and resources. Traditional gender roles, particularly in the German-speaking population, might affect how men and women perceive and report physical limitations, amplifying sex-based differences in frailty prevalence. These sociocultural and systemic factors highlight the need to contextualise frailty-screening tools within specific populations. Addressing these nuances is crucial to ensure that tools like the PRISMA-6 reduce inherent biases and adapt to diverse cultural and societal landscapes.

These findings provide substantial evidence to support the formal validation of the PRISMA-6 as an independent frailty-screening instrument. This validation process should encompass an evaluation of the PRISMA-6 and PRISMA-5 (excluding both item 2 and item 6, which refers to the possibility of having a person that may provide help in case of need, as proposed in the Greek validation study [20]) in comparison with established instruments such as the Clinical Frailty Scale [30]. The analysis should assess their capacity to predict adverse outcomes, including hospitalisation and functional decline, while examining internal consistency, reliability, and construct validity to determine if the exclusion of these items enhances the coherence of the instruments. Moreover, the validation process should ensure that the PRISMA-6 and PRISMA-5 perform consistently across sex without compromising sensitivity or specificity, thereby addressing potential sex and cultural biases.

## 5. Conclusions

This investigation elucidates the limitations of the PRISMA-7 as a frailty-screening instrument attributable to the sex bias introduced by item 2. The PRISMA-6 is a promising alternative, demonstrating greater congruence with established frailty patterns and reduced sex bias. Given the increasing emphasis on population-level frailty screening, validating the PRISMA-6 could significantly contribute to equitable and effective healthcare planning and resource allocation. Subsequent research should prioritise this validation to ensure robust and unbiased frailty assessment in diverse clinical and community settings.

## Figures and Tables

**Figure 1 geriatrics-10-00009-f001:**
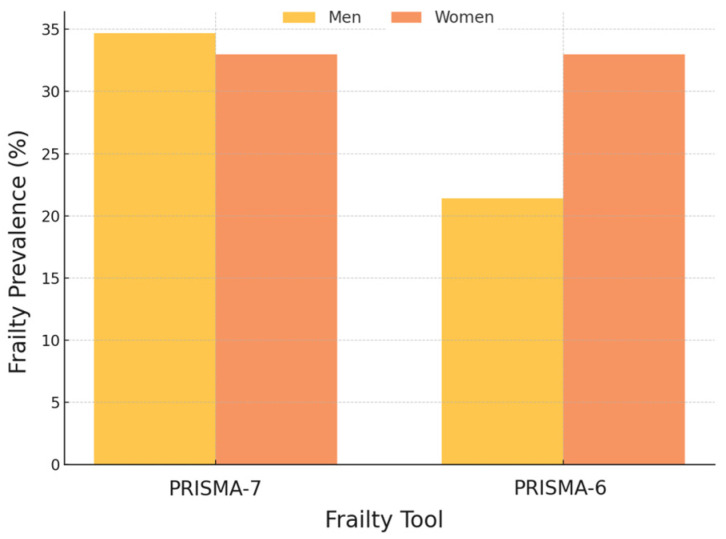
Gender differences in frailty prevalence by the PRISMA-7 and PRISMA-6 rates (defined as a score of 3 or higher).

**Table 1 geriatrics-10-00009-t001:** Sample characteristics and prevalence of the PRISMA-7 frailty.

Characteristics	Total (*n* = 1695)	Non-frail (*n* = 1121)	Frail (*n* = 574)	*p*-Value ^1^
Sex, % (*n*)				0.50
Female	48.1 (815)	67.0 (546)	33.0 (269)	
Male	51.9 (880)	65.3 (575)	34.7 (305)	
Age (years), % (*n*)				
75–84 years	59.3 (1005)	86.3 (867)	13.7 (138)	<0.001
≥85 years	40.7 (690)	36.8 (254)	63.2 (436)	
Native tongue, % (*n*)				0.12
German	53.2 (902)	64.4 (581)	35.6 (321)	
Italian	42.1 (713)	68.7 (490)	31.3 (223)	
Ladin	4.0 (67)	65.7 (44)	34.3 (23)	
Other	0.8 (13)	46.2 (6)	53.8 (7)	
Citizenship, % (*n*)				0.17
Italian	98.7 (1673)	66.3 (1110)	33.7 (563)	
Other	1.3 (22)	50.0 (11)	50.0 (11)	
Community, % (*n*)				0.05
Urban	45.6 (773)	68.2 (629)	31.8 (293)	
Rural	54.4 (992)	63.6 (492)	36.4 (281)	
Living situation, % (*n*)				0.23
Living alone	30.4 (516)	77.1 (791)	32.9 (388)	
Living with partner/family	69.6 (1179)	64.0 (330)	36.0 (186)	
Children, % (*n*)				0.05
Yes	79.3 (1344)	65.0 (873)	35.0 (471)	
No	20.7 (351)	70.7 (248)	29.3 (103)	
Educational level, % (*n*)				<.001
Below high school	77.8 (1319)	63.3 (835)	36.7 (484)	
High school or higher	22.2 (376)	76.1 (286)	23.9 (90)	
Financial resources, % (*n*)				<0.001
Excellent or good	28.5 (483)	73.1 (353)	26.9 (130)	
Adequate	49.4 (837)	67.7 (567)	32.3 (270)	
Insufficient or low	22.1 (375)	53.6 (201)	46.4 (174)	
Overall optimism, % (*n*)				<0.001
Yes	84.5 (1433)	69.9 (119)	30.1 (143)	
No	16.5 (262)	45.4 (1002)	54.6 (431)	
Health status, % (*n*)				<0.001
Poor or moderate	59.7 (1012)	50.7 (513)	49.3 (499)	
Good or very good	40.3 (683)	89.0 (608)	11.0 (75)	
Physical activity				<0.001
2 h or more a week	43.2 (733)	84.4 (619)	15.6 (114)	
Less than 2 h a week	40.1 (679)	59.6 (405)	40.4 (274)	
Never	16.7 (283)	34.3 (97)	65.7 (186)	

^1^ *p*-value, chi-squared test.

**Table 2 geriatrics-10-00009-t002:** Sample characteristics and prevalence of the PRISMA-6 frailty.

Characteristics	Total Sample (*n* = 1695)	Non-Frail (*n* = 1238)	Frail (*n* = 457)	*p*-Value ^1^
Sex, % (*n*)				<0.001
Female	48.1 (815)	67.0 (692)	33.0 (188)	
Male	51.9 (880)	78.6 (546)	21.4 (269)	
Age (years), % (*n*)				<0.001
75–84 years	59.3 (1005)	89.9 (903)	10.1 (102)	
≥85 years	40.7 (690)	48.6 (335)	51.4 (355)	
Native tongue, % (*n*)				0.06
German	53.2 (902)	71.7 (647)	28.3 (255)	
Italian	42.1 (713)	75.3 (537)	24.7 (176)	
Ladin	4.0 (67)	71.6 (48)	28.4 (19)	
Other	0.8 (13)	46.2 (6)	53.8 (7)	
Citizenship, % (*n*)				0.08
Italian	98.7 (1673)	73.3 (1226)	26.7 (447)	
Other	1.3 (22)	54.5 (12)	45.5 (10)	
Community, % (*n*)				0.03
Urban	45.6 (773)	75.3 (694)	24.7 (228)	
Rural	54.4 (992)	70.4 (544)	29.6 (229)	
Living situation, % (*n*)				0.02
Living alone	30.4 (516)	74.7 (881)	25.3 (298)	
Living with partner/family	69.6 (1179)	69.2 (357)	30.8 (159)	
Children, % (*n*)				0.06
Yes	79.3 (1344)	71.9 (967)	28.1 (377)	
No	20.7 (351)	77.2 (271)	22.8 (80)	
Educational level, % (*n*)				<0.001
Below high school	77.8 (1319)	80.0 (923	30.0 (396)	
High school or higher	22.2 (376)	83.8 (315)	16.2 (61)	
Financial resources, % (*n*)				<0.001
Excellent or good	28.5 (483)	81.4 (393)	18.6 (90)	
Adequate	49.4 (837)	75.0 (628)	25.0 (209)	
Insufficient or low	22.1 (375)	57.9 (217)	42.1 (158)	
Overall optimism, % (*n*)				<0.001
Yes	84.5 (1433)	53.8 (141)	46.2 (121)	
No	16.5 (262)	76.6 (1097)	23.4 (336)	
Health status, % (*n*)				<0.001
Poor or moderate	59.7 (1012)	58.9 (596)	41.1 (416)	
Good or very good	40.3 (683)	94.0 (642)	6.0 (41)	
Physical activity				<0.001
2 h or more a week	43.2 (733)	90.6 (664)	9.4 (69)	
Less than 2 h a week	40.1 (679)	68.2 (463)	31.8 (216)	
Never	16.7 (283)	39.2 (111)	60.8 (172)	

^1^ *p*-value, chi-squared test.

**Table 3 geriatrics-10-00009-t003:** Sex differences in the PRISMA-6 frailty prevalence.

Characteristics	Frail (*n* = 457)	Male (*n* = 188)	Female (*n* = 269)	*p*-Value ^1^
Age (years), % (*n*)				0.09
75–84 years	10.1 (102)	18.1 (34)	25.3 (68)	
≥85 years	51.4 (355)	81.9 (154)	74.7 (201)	
Native tongue, % (*n*)				0.04
German	28.3 (255)	50.5 (95)	59.5 (160)	
Italian	24.7 (176)	45.7 (86)	33.5 (90)	
Ladin	28.4 (19)	2.7 (5)	5.2 (14)	
Other	53.8 (7)	1.1 (2)	1.9 (5)	
Citizenship, % (*n*)				1.00
Italian	26.7 (447)	97.9 (184)	97.8 (263)	
Other	45.5 (10)	4 (2.1)	2.2 (6)	
Community, % (*n*)				0.28
Rural	29.6 (229)	46.8 (88)	52.4 (141)	
Urban	24.7 (228)	53.2 (100)	47.6 (128)	
Living situation, % (*n*)				<0.001
Living alone	25.3 (298)	80.3 (151)	54.6 (147)	
Living with partner/family	30.8 (159)	19.7 (37)	45.4 (122)	
Children, % (*n*)				0.16
Yes	28.1 (377)	79.3 (149)	84.8 (228)	
No	22.8 (80)	20.7 (39)	15.2 (41)	
Educational level, % (*n*)				0.07
Below high school	30.0 (396)	83.0 (156)	89.2 (240)	
High school or higher	16.2 (61)	17.0 (32)	10.8 (29)	
Financial resources, % (*n*)				0.01
Excellent or good	18.6 (90)	22.9 (43)	47 (17.5)	
Adequate	25.0 (209)	50.5 (95)	42.4 (114)	
Insufficient or low	42.1 (158)	26.6 (50)	40.1 (108)	
Overall optimism, % (*n*)				0.36
Yes	46.2 (121)	23.9 (45)	28.3 (76)	
No	23.4 (336)	76.1 (143)	71.7 (193)	
Health status, % (*n*)				0.83
Poor or moderate	41.1 (416)	90.4 (170)	91.4 (246)	
Good or very good	6.0 (41)	9.6 (18)	8.6 (23)	
Physical activity				0.08
2 h or more a week	9.4 (69)	18.6 (35)	12.6 (34)	
Less than 2 h a week	31.8 (216)	48.9 (92)	46.1 (124)	
Never	60.8 (172)	32.4 (61)	41.3 (111)	

^1^ *p*-value, chi-squared test.

**Table 4 geriatrics-10-00009-t004:** Factors explaining frailty in community-dwelling older adults (*n* = 1695).

	PRISMA-7			PRISMA-6		
	OR	95% CI	*p*-Value ^1^	OR	95% CI	*p*-Value ^1^
Age and sex						
Females (reference group: males)	0.42	0.31; 0.57	<0.001	1.34	0.98; 1.83	0.067
Age ≥ 85 years (reference group: 75–84 years)	12.6	9.45; 17.0	<0.001	9.87	7.30; 13.5	<0.001
Native tongue (reference group: German)						
Italian	0.62	0.43; 0.88	0.008	0.77	0.53; 1.12	0.2
Ladin and others	1.03	0.53; 1.98	>0.9	1.24	0.61; 2.46	0.5
Rural community (reference group: urban)	0.87	0.61; 1.23	0.4	1.01	0.70; 1.45	>0.9
Living alone (reference group: living with family)	1.02	0.75; 1.40	0.9	0.97	0.70; 1.33	0.8
Children (reference group: yes)	0.85	0.60; 1.21	0.4	0.89	0.61; 1.28	0.5
Educational level (reference group: below high school)	0.90	0.62; 1.31	0.6	0.93	0.62; 1.39	0.7
Financial resources (reference group: excellent or good)						
Adequate	1.21	0.86; 1.71	0.3	1.28	0.88; 1.87	0.2
Insufficient or low	1.68	1.12; 2.52	0.012	2.07	1.36; 3.16	<0.001
Lifestyle and health-related factors						
Overall optimism (reference group: no)	0.47	0.33; 0.67	<0.001	0.54	0.38; 0.78	<0.001
Good or very good health status (reference group: poor or moderate)	0.12	0.09; 0.17	<0.001	0.11	0.07; 0.16	<0.001
Physical activity (reference group: ≥2 h a week)						
Less than 2 h a week	2.60	1.89; 3.59	<0.001	2.64	1.86; 3.77	<0.001
Never	6.67	4.46; 10.1	<0.001	7.58	5.01; 11.6	<0.001

^1^ *p*-value, logistic regression analyses. Abbreviations: OR, odds ratio; CI, confidence interval.

## Data Availability

Research data are available from the corresponding author upon reasonable request.

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
