# Peer review of "Comparing the PRISMA-7 and a Modified Version (PRISMA-6) for Frailty Screening: Addressing Sex Bias in Community-Dwelling Older Adults"

_geriatrics, 2025, doi:10.3390/geriatrics10010009_

Round 1
Reviewer 1 Report
Comments and Suggestions for Authors
The authors are advised to address the following comments to enhance the quality of the manuscript.
The manuscript employs the terms "gender" and "sex" interchangeably; however, it is recommended that the authors employ a single term consistently.
The title is excessively detailed and could be streamlined. Moreover, the central objective of this study is to compare PRISMA 7 with PRISMA 6, a fact that must be reflected in the title.
While the abstract is structured, it is verbose and overly technical. To enhance its accessibility, the language should be simplified. For instance, phrases such as "providing evidence-based recommendations for its implementation" should be replaced with more accessible language, such as "offering practical recommendations." Additionally, the implications of PRISMA-6's enhanced internal consistency are underemphasized.
The discussion on PRISMA-7's limitations is introduced in the Introduction only after a considerable delay. This argument would be more compelling if it were introduced earlier in the text. Additionally, the potential influence of cultural and societal variations on frailty outcomes is only superficially introduced and lacks a thorough exploration.
In the Method section, while the rationale behind excluding item 2 is adequately delineated, there is a paucity of discussion regarding the justification for its inclusion in the original tool.
The incorporation of graphs or charts (e.g., bar charts illustrating frailty prevalence by gender and tool) would considerably enhance comprehension and engagement.
The results could have been strengthened by subgroup analyses (e.g., by age, socioeconomic status, or rural-urban location) to uncover patterns masked by aggregate data.
The claim that PRISMA-6 is more "equitable" lacks robust evidence beyond internal consistency improvements and prevalence shifts. Additional validation is necessary. Or at least the authors need to explain why the increase of the internal consistency is sufficient to point out PRISMA 6 is better than PRISMA 7. Plus, the broader implications of PRISMA-6’s gender-neutral approach on frailty interventions (e.g., resource allocation, clinical accuracy) are not fully addressed.
The discussion alludes to the unique sociocultural context of South Tyrol, yet it does not thoroughly explore how this context might influence the results, such as through language differences, access to care, or gender norms.
Reviewer 2 Report
Comments and Suggestions for Authors
Using diagnostic tools in a reliable and credible way to assess the phenomenon being studied is extremely important. This is becoming an indispensable requirement for diagnostic tools and tests used in everyday clinical practice. Therefore, the topic taken up by the authors of the reviewed work is, in my opinion, crucial, in the light of new legislative solutions concerning the care of older patients in primary health care. It is important that the tools used to assess patients for the presence or absence of frailty syndrome are reliable and carry the appropriate information load, since diagnostic and therapeutic interventions are to follow this.
The work is written correctly, although it was not possible to avoid certain imperfections, which, due to my role as a reviewer, I will allow myself to point out below.
1/ It is a pity that another instrument for assessing frailty syndrome, insensitive to gender influences, was not included - the authors themselves indicate this fact as one of the weak points of the study, which must be agreed with.
2/ A limitation is certainly the reliance on self-assessment. Therefore, the lack of including, for example, information on cognitive impairment may also be a limitation and affect the results in the group of respondents aged 75+.
3/ What was treated as "Physical activity"? This should be described in the methodology.
4/ A large range between the number of randomly selected - 3600 and those who took part - 1695 (response rate = 47.08%) - there is no explanation of who did not respond (age / gender / place of residence). It is worth briefly referring to this, because it could also have affected the obtained results - was the sample really representative?
5/ It may be worth assessing the agreement of the Prisma-6 and Prisma 7 classifications using cohen's Kappa.
Round 2
Reviewer 1 Report
Comments and Suggestions for Authors
The authors have addressed the comments, and the manuscript has been improved sufficiently for its publication. I thank the authors for their efforts on this manuscript.
Reviewer 2 Report
Comments and Suggestions for Authors
The authors have addressed my comments sufficiently, including them in the manuscript. I believe the manuscript deserves publication.